

# Heterogeneity of the effect of the COVID-19 pandemic on the incidence of Metabolic Syndrome onset at a Japanese campus

Toshiharu Mitsuhashi

Center for Innovative Clinical Medicine, Okayama University Hospital, Okayama, Okayama Prefecture, Japan

Corresponding author
Toshiharu Mitsuhashi,
mitsuh-t@cc.okayama-u.ac.jp

## ABSTRACT

**Background**. The coronavirus disease 2019 (COVID-19) outbreak began in China in December 2019, with the World Health Organization declaring a state of emergency in January 2020. Worldwide implementation of lockdown measures to slow the spread of the virus led to reduced physical activity, disrupted eating habits, mental health issues, and sleep disturbances, which increased the risk of lifestyle-related diseases such as metabolic syndrome (MetS). During the COVID-19 pandemic, healthcare workers, especially intensive care workers, experienced longer working hours and burnout, which further increased the risk of lifestyle-related diseases. Accordingly, it is important to identify individuals at a risk of new-onset MetS during a pandemic, which could direct preventive interventions. This study aimed to assess the heterogeneous impact of the COVID-19 pandemic on the incidence of new-onset MetS based on the conditional average treatment effect (CATE) and to identify at-risk populations.
**Methods**. This study analyzed health checkup data obtained from Okayama University Shikata Campus workers using paired baseline and follow-up years. Baseline data encompassed 2017 to 2019, with respective follow-up data from 2018 to 2020. Furthermore, as the COVID-19 pandemic in Japan began in January 2020, workers who underwent follow-up health checkups in 2018 to 2019 and 2020 were considered as "unexposed" and "exposed," respectively. As the Shikata campus has several departments, comparisons among departments were made. The primary outcome was new-onset MetS at follow-up. Predictor variables included baseline health checkup results, sex, age, and department (administrative, research, medical, or intensive care department). X-learner was used to calculate the CATE.
**Results**. This study included 3,572 eligible individuals (unexposed, $n = 2,181$; exposed, $n = 1,391$). Among them, 1,544 (70.8%) and 866 (62.3%) participants in the unexposed and exposed groups, respectively, were females. The mean age ($\pm$standard deviation) of the unexposed and exposed groups was $48.2 \pm 8.2$ and $47.8 \pm 8.3$ years, respectively. The COVID-19 pandemic increased the average probability of new-onset MetS by 4.4% in the overall population. According to the department, the intensive care department showed the highest CATE, with a 15.4% increase. Moreover, there was large heterogeneity according to the department. The high-CATE group was characterized by older age, urinary protein, elevated liver enzymes, higher triglyceride levels, and a history of hyperlipidemia treatment.
**Conclusions**. This study demonstrated that the COVID-19 pandemic increased the incidence of new-onset MetS, with this effect showing heterogeneity at a single Japanese

campus. Regarding specific populations, workers in the intensive care department showed an increased risk of new-onset MetS. At-risk populations require specific preventive interventions in case the current COVID-19 pandemic persists or a new pandemic occurs.

# INTRODUCTION

In December 2019, a patient exhibiting symptoms of viral pneumonia was reported in Wuhan, Hubei Province, China (*Chen et al., 2020*; *Huang et al., 2020*). In January 2020, the causative agent was identified as a novel coronavirus, severe acute respiratory syndrome coronavirus 2, which causes coronavirus disease 2019 (COVID-19). Owing to the subsequent rapid spread of the virus, the World Health Organization declared a Public Health Emergency of International Concern on January 30, 2020 (*Eurosurveillance Editorial Team, 2020*). Accordingly, lockdown measures for slowing the transmission of COVID-19 were implemented globally. Similarly, Japan declared a state of emergency, urging residents to stay at home, temporarily closing businesses and schools, and requesting travelers to postpone their plans (*Looi, 2020a*; *Looi, 2020b*; *Aoki, 2021*; *Watanabe & Yabu, 2021*).

During the COVID-19 pandemic, there has been an increased risk of lifestyle-related diseases (*Lim, Kong & Tuomilehto, 2021*). This could be attributed to decreased physical activity due to restricted outdoor movements and closures of gyms and sports facilities (*Martinez-Ferran et al., 2020*; *Wilke et al., 2021*; *Mehraeen et al., 2023*). Moreover, there has been an increasing trend of the body mass index due to the disruption of eating habits (*Martinez-Ferran et al., 2020*; *Clemente-Suárez et al., 2021*; *Akter et al., 2022*). Furthermore, concerns regarding severe acute respiratory syndrome coronavirus 2 infection and stress resulting from self-restraint have contributed to widespread mental health issues and sleep disturbances (*Pfefferbaum & North, 2020*; *Xiong et al., 2020*; *Jahrami et al., 2021*). This also contributes to the increased risk of lifestyle-related diseases (*Clemmensen, Petersen & Sørensen, 2020*). Additionally, individuals undergoing treatment for lifestyle-related diseases experienced reduced access to medical care (*Seidu, Sk & Khunti, 2021*; *Maeda et al., 2022*; *Yagome et al., 2022*).

In understanding the context of the health impact caused by the COVID-19 pandemic, it is crucial to investigate metabolic syndrome (MetS), a cluster of conditions including increased blood pressure, high blood sugar levels, excess body fat around the waist, and abnormal cholesterol or triglyceride levels, which together increase the risk of heart diseases, stroke, and type 2 diabetes (*Mohammad, 2018*). The relevance of MetS in the current global health landscape is underscored by its rising prevalence, which is intimately linked to lifestyle factors such as physical inactivity, a poor diet, and obesity. The COVID-19 pandemic, with its unprecedented impact on daily life, has potentially exacerbated these lifestyle-related risk factors (*Auriemma et al., 2021*). Consequently, the pandemic
generated a unique environment that increased the risk of MetS, particularly in populations undergoing significant lifestyle changes due to COVID-19 pandemic-related restrictions (*Yanai, 2020*; *Dissanayake, 2023*).

This increased risk of lifestyle-related diseases during the COVID-19 pandemic is of particular concern for healthcare workers, especially those working in intensive care departments. In the early stages of the pandemic, healthcare workers experienced burnout due to extended working hours and the urgency of responding to emergencies (*George et al., 2020*; *Lee & Lee, 2020*; *Søvold et al., 2021*). This can result in reduced physical activity (*Is et al., 2021*; *Kua et al., 2022*) and altered dietary habits (*Yu et al., 2021*; *Yaman & Hocaoğlu, 2023*). Additionally, these factors contribute to mental health problems and sleep disturbances (*Pappa et al., 2020*; *Stewart et al., 2021*), especially among frontline intensive care workers (*Gupta & Sahoo, 2020*; *Koontalay et al., 2021*).

As exposure to the COVID-19 pandemic is inevitable, it is important to identify populations vulnerable to the increased risk of lifestyle-related diseases and to implement targeted preventive interventions. The aforementioned findings suggest that the impact of the COVID-19 pandemic on lifestyle-related diseases could be heterogeneous. Elucidating the heterogeneity of this effect may provide insights into prevention targets and strategies (*Van Der Weele & Knol, 2014*; *Greifer & Stuart, 2021*). Nevertheless, to the best of our knowledge, there has been no comprehensive study on the heterogeneity of the impact of the COVID-19 pandemic on lifestyle-related diseases.

Therefore, this study aimed to assess the heterogeneity of the impact of the COVID-19 pandemic on the incidence of new-onset metabolic syndrome (MetS) among university campus workers.

## MATERIALS & METHODS

### Study design
This was an observational study that utilized previously collected data.

### Data source and participants
This study included data obtained from workers at the Shikata Campus of Okayama University. As this campus has various departments, comparisons were performed among the departments. Although particular attention was given to medical workers, data from all departments were used for improved elucidation of the characteristics of the high-risk group and the accuracy of the causal inference model. The dataset was provided by the Junpukai healthcare center, which conducts health checkups at the campus.

All participants met the following inclusion criterion: workers who underwent health checkups for at least two consecutive years from 2017 to 2020. The following individuals were excluded from the study: (1) individuals who only had data unrelated to general health assessments, (2) individuals who were not tested for MetS, and (3) individuals with MetS at baseline. Health checkup data obtained in 2017, 2018, and 2019 were the baseline data with the respective follow-ups in 2018, 2019, and 2020. All data that were obtained for this study met the selection criteria.

As this study only used pre-existing information, an *a priori* sample size calculation was performed. To ensure the reproducibility of the analysis, the dataset and analysis code were made publicly available. However, to protect personal information, personal identifiers were removed; further, quasi-identifiers were anonymized by resampling and randomization. Anonymized data were used for analysis in this study. However, subtle discrepancies emerged between the original and anonymized datasets during the anonymization process. Although privacy concerns preclude the disclosure of the original dataset, a comparison was conducted to ascertain any differences in the analytical results between the two datasets.

### Outcome variable

Follow-up data were used to determine whether participants had MetS based on the diagnostic criteria provided by the Ministry of Health, Labour and Welfare (MHLW) (short URL: https://bit.ly/3Jx9b8g).

Based on Japan's MetS diagnostic criteria (*Journal of the Japanese Society of Internal Medicine, 2005*), a flowchart was crafted from health checkup data of workers, considering challenges such as missing data and non-fasting blood glucose levels. It aims to provide accurate diagnoses under these constraints, hence its complexity. Differences from international criteria (*Alberti et al., 2009*) are detailed in Article S1, with the English flowchart.

Both Japanese and international criteria use similar tests, but with different cutoffs: high-density lipoprotein (HDL) cholesterol (40 mg/dL), triglycerides (150 mg/dL), fasting blood sugar (110 mg/dL), and blood pressure (130/85 mmHg) for Japan. The international criteria have sex-specific HDL thresholds, whereas Japan has uniform thresholds. Additionally, Japan combines triglyceride and HDL cholesterol as one item, unlike the international approach. Japan's cutoff for blood glucose differs from the international standard (100 mg/dL), and they use different abdominal circumference measurement points. Internationally, it is midway between the lower rib margin and the iliac crest; in Japan, it is at the umbilicus level. This alters cutoffs: internationally, 90 cm (males) and 80 cm (females); in Japan, 85 cm (males) and 90 cm (females). Japan defines MetS as having excess visceral fat, indicated by abdominal circumference, plus two or more cardiovascular risk factors (hyperglycemia, dyslipidemia, high blood pressure).

### Exposure variable

The exposure was defined as the COVID-19 pandemic. As the COVID-19 pandemic in Japan began in January 2020, workers who underwent follow-up health checkups in 2018–2019 and 2020 were considered as "unexposed" and "exposed", respectively. The definition of exposure status is more clearly shown in Fig. 1.

### Predictive variables

Health-checkup items, sex, age, and working department obtained during baseline health checkups were used as predictor variables. Health-checkup items are summarized in Article S2. Missing continuous variables were imputed using median values; indicator variables were used to indicate missing data. Missing categorical variables were assigned to

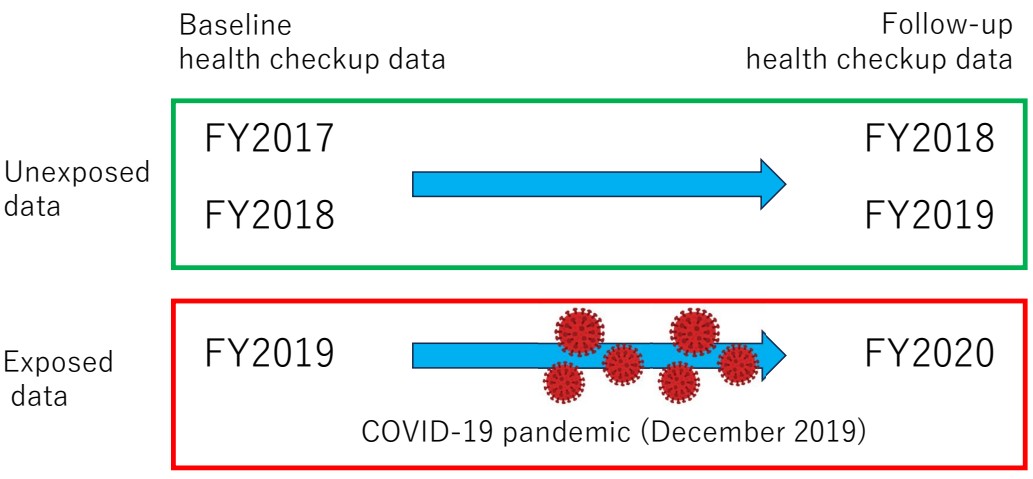

**Figure 1** **Definition of exposed and unexposed data.** FY: fiscal year.

a separate missing category. Departments were divided into four categories: administrative (library, cafeteria, *etc.*), research, medical (excluding the intensive care department), and intensive care department.

## Statistical analysis

Descriptive statistics were used to analyze baseline health checkup data. Continuous variables are presented as means with standard deviations, while categorical variables are presented as frequencies and percentages. Missing values were excluded during the computation of descriptive statistics.

To assess the impact of the COVID-19 pandemic on the incidence of new-onset MetS, the conditional average treatment effect (CATE) was calculated. CATE is defined as a difference between the expected outcomes of two treatments based on covariates, representing an individualized causal effect in causal inference (*Abrevaya, Hsu & Lieli, 2015*; *Jacob, 2021*; *Kato & Imaizumi, 2023*). CATE for the covariate $X = x$ is defined using the flowing formula (*Nie & Wager, 2021*). Note that $Y^1$ and $Y^0$ are the potential outcomes of exposure and non-exposure, respectively.

$$\text{CATE}(x) = E(Y^1 - Y^0 | X = x).$$

The main benefit of using CATE is its ability to uncover heterogeneity in treatment effects across subpopulations, which is crucial for personalized treatments and decision-making in various domains such as economics and healthcare (*Jacob, 2021*). In contrast to ATE, which estimates an average treatment effect for the entire population, CATE provides a more detailed understanding by estimating the effect specifically for subgroups within the population, thereby addressing the heterogeneity that ATE may overlook. For example, in economics, research with the advantages of CATE has been conducted (*Crépon et al., 2015*).

Traditionally, stratified analyses and analyses incorporating interaction terms have been employed to calculate the CATE. Contrastingly, this study utilized X-learner, which

is a machine learning-based causal inference method that allows more flexible CATE calculation (*Künzel et al., 2019a*; *Künzel et al., 2019b*; *Jacob, 2021*). Furthermore, 95% confidence intervals (95% CIs) for CATE were computed using the bootstrap method (15,000 iterations) (*Künzel et al., 2019b*).

Values of CATE were calculated for the overall population, department, sex, and age. The difference in CATE was calculated for each category (division and sex). For division, I calculated the difference between the medical division and the other three divisions, using the medical division as a control. For sex, I calculated the difference between the females and the males, using the female as a control. Scatter plots were generated to illustrate the association between the CATE and age. Approximate curves with locally weighted regression smoothing were applied to the scatterplot to demonstrate the heterogeneity of the association between CATE and age according to the department. The population characteristics with CATE values in the upper and lower 10th percentiles were compared, and the standardized differences were calculated. All statistical analyses were performed using Stata 17/MP8 (StataCorp LLC, College Station, TX, USA). Statistical significance was set at $p < 0.05$.

## Machine learning model: X-learner

Figure 2 presents a schematic representation of the X-learner method. Figure 2 was created based on the website (*Matheus, 2023*). First, models were constructed to predict outcomes using predictive variables in the exposed and non-exposed subgroups. The logistic regression model was used as the base learners M0 and M1, as the outcome was binary, and the computational load was very low. The area under the curve (AUC) was used to assess predictive utility; subsequently, the set of predictive variables with the highest AUC values in the 10-fold cross-validation was selected. Sets of predictive variables were selected from the original variables, squared variables, and interaction terms (Article S3).

Next, the imputed treatment effect (ITE) was calculated using the difference between the predicted and observed outcomes for the exposed and unexposed subgroups. Accordingly, there were no unexposed and exposed ITEs in the exposed and unexposed groups, respectively.

Subsequently, models were constructed to predict the ITE using predictive variables in the exposed and non-exposed subgroups (base learners: M2 and M3 in Fig. 2). The response variable in this prediction model, ITE, was assigned a real number in the range from −1 (100% prevention of new-onset MetS after exposure) to 1 (100% occurrence of new-onset MetS after exposure). As there are no appropriate regression models for this ITE, a beta regression model was used after transforming the data by adding 1 to the original value and dividing by 2 (new range: 0–1). The root mean squared error (RMSE) was used to evaluate the predictive utility; additionally, the set of predictive variables with the lowest value in the 10-fold cross-validation was selected (Article S3).

Finally, the weighted average of the ITE predictions (CATE1 and CATE0 in Fig. 2) obtained from M2 and M3 was used to estimate the CATE. Conditional exposure probabilities (*i.e.,* propensity scores), which reduce bias through simulations (*Künzel et al., 2019a*; *Künzel et al., 2019b*), are usually used for weighting. However, as the exposure

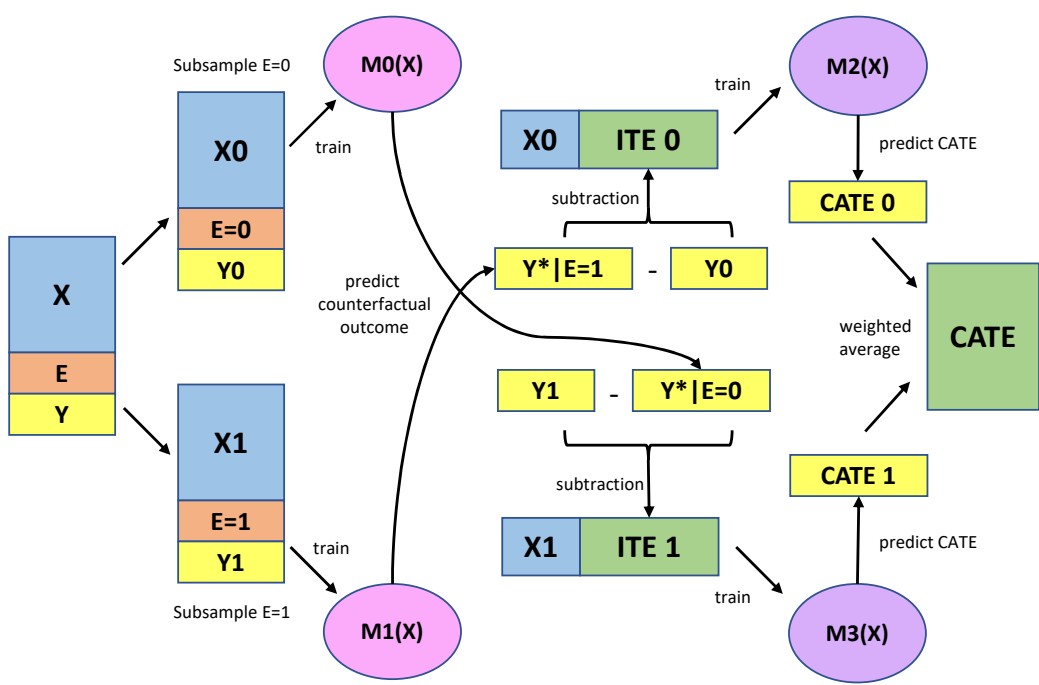

**Figure 2** **Schema diagram for X-learner.** X, Predictive variables; E, Exposure variable; Y, Outcome variable; M0 (X), Logistic regression model for predicting the outcome constructed using unexposed subsamples; M1 (X), Logistic regression model for predicting the outcome constructed using exposed subsamples; ITE, Imputed treatment effect; M2 (X), Beta regression model for predicting ITE constructed using the unexposed subsample; M3 (X), Beta regression model for predicting ITE constructed using the exposed subsample; CATE, Conditional average treatment effect.

variable in this study was independent of covariates, *i.e.,* health checkup results did not influence whether an individual experienced the COVID-19 pandemic, marginal exposure probability was applied for weighing.

This study used the following three assumptions to apply the X-learner approach for causal inference. First, it is assumed that enough covariates have been measured to calculate potential outcomes from these covariates accurately. Second, it is postulated that potential outcomes remain consistent before and after the COVID-19 pandemic, given the covariates. Furthermore, in calculating the ITE from the covariates, there is an assumption that an adequate number of covariates are measured to ensure precise calculation of ITE.

## Ethical approval

This study was ethically approved by the clinical research review committee of the Junpukai healthcare center (approval date: June 3, 2022, approval number: 20210015). As this study only used pre-existing information and the data had been anonymized, informed consent had already been obtained through information disclosure and opt-out provisions negating the need to obtain consent from each individual.

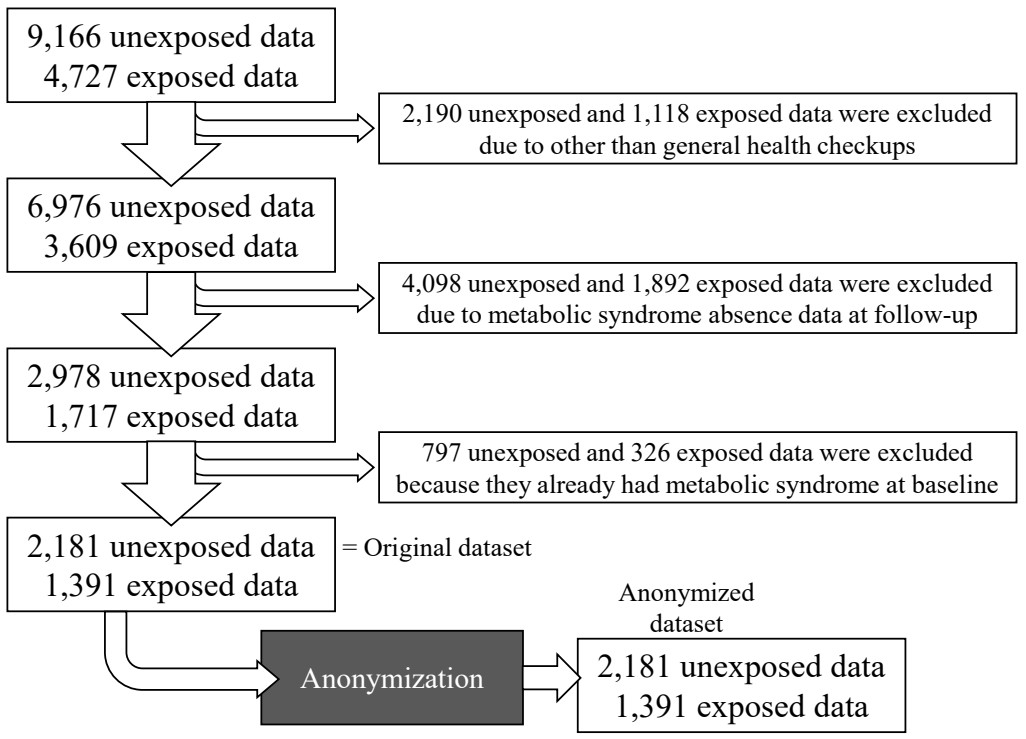

**Figure 3  Flowchart of health checkup data in the study.** The data analysis and results are based on anonymized data. Anonymization was performed through resampling and random number addition, which impeded identification of an individual from a pseudo-identifier.

## RESULTS

### Descriptive statistics

The dataset comprised data obtained from 13,993 health checkups. After excluding irrelevant data, data with missing outcomes, and data showing MetS at baseline, the total number of records was 3,572 (non-exposed, $n = 2,181$; exposed, $n = 1,391$). Subsequently, the data were anonymized and analyzed (Fig. 3).

In the original data, 629 individuals had only one measurement, 612 had two, and 2,331 had three; however, the same person could not be identified in the anonymized data used for analysis, which should generally be avoided with respect to independent and identically distributed variables. Despite this situation, the overall impact of the violation of independence assumption is mitigated by several factors. Firstly, no individual had repeated measures in the exposed data subset, thus upholding the independence assumption in estimating the models m1 and m3 depicted in Fig. 2. Conversely, some individuals had up to two measurements in the unexposed data subset, suggesting a potential breach of the independence assumption. However, the effect of this violation is minimal. This is primarily because the anonymization process involved adding random numbers to the variables, reducing the correlation within individuals' repeated measurements. Thus, although the original data might have breached the independence assumption, the analyzed anonymized

data largely preserves this principle. Additionally, the main objective of the analysis was not to derive regression coefficients but to develop outcome prediction models (as shown in m0 and m2 of Fig. 2). These models were evaluated using 10-fold cross-validation AUC and RMSE metrics, demonstrating non-poor performance. Furthermore, the sensitivity analysis also confirmed that the effect of violating the independence assumption was not substantial (Article S3). Therefore, any potential violation of the independence assumption has a limited effect.

Table 1 presents the descriptive statistics results. Most participants were females (1,544 (70.8%) and 866 (62.3%) in the unexposed and exposed groups, respectively). The mean age ($\pm$standard deviation) was 48.2 $\pm$ 8.2 and 47.8 $\pm$ 8.3 years in the unexposed and exposed groups, respectively. There was a slight difference in the incidence of new-onset MetS during the follow-up period between the unexposed ($n = 66$ (3.0%)) and exposed ($n = 72$ (5.2%)) groups, respectively.

There was no significant between-group difference in the baseline urinalysis and blood test results (Table 2).

## Evaluation of base learners

According to the logistic regression model, the AUC values for M0 and M1 were 0.95055 and 0.89743, respectively. According to the beta regression model, the RMSE values for M2 and M3 were 0.10096 and 0.14933, respectively (Article S3).

Model M0 exhibited outstanding discrimination with an AUC exceeding 0.9, while model M1 demonstrated excellent discrimination, having an AUC above 0.8, in line with Hosmer and Lemeshow's criteria (*Hosmer Jr, Lemeshow & Sturdivant, 2013*).

For models M2 and M3, RMSE values ranged from 0 (complete accuracy) to 1 (complete inaccuracy), reflecting real-number outcomes between 0 and 1 in a beta regression model. The RMSE results revealed an average error of approximately 10–15%. Despite the absence of a standard benchmark for RMSE interpretation, these values were deemed acceptable, not significantly undermining the study's validity, and thus, the research was continued.

## CATE estimates

As shown in Table 3, the CATE estimate for the overall population ($n = 3,572$) was 0.044 (95% CI [0.008–0.080]; $p = 0.017$). This indicates that the COVID-19 pandemic increased the probability of new-onset MetS by 4.4%. Figure 4 shows a histogram depicting this distribution, with CATEs being widely distributed but most frequently occurring around 0 (*i.e.,* no causal effect). However, 389 (10.4%) participants had a CATE > 0.25.

According to the department, the CATE values in the administrative, research, medical, and intensive care departments were 0.034 (95% CI [−0.025–0.094]; $p = 0.256$), 0.068 (95% CI [−0.016–0.153]; $p = 0.112$), 0.032 (95% CI [−0.012–0.077]; $p = 0.158$), and 0.154 (95% CI [−0.021–0.329]; $p = 0.084$), respectively. This indicated heterogeneity in the impact, with the intensive care department having a particularly high CATE.

Regarding sex, the CATE values for female and male workers were 0.033 (95% CI [0.005–0.062]; $p = 0.022$) and 0.066 (95% CI [−0.016–0.148]; $p = 0.116$), respectively, indicating heterogeneity between sexes.

**Table 1   Descriptive statistics of the baseline health-checkup results (1).**

| | | Before COVID (N = 2,181) | After COVID (N = 1391) |
|---|---|---|---|
| Sex | Female | 1544 (70.8%) | 866 (62.3%) |
| | Male | 637 (29.2%) | 525 (37.7%) |
| Age | Mean ± SD | 48.2 ± 8.2 (n = 2,181) | 47.8 ± 8.3 (n = 1,391) |
| Division | Administrative | 535 (24.5%) | 302 (21.7%) |
| | Research | 260 (11.9%) | 175 (12.6%) |
| | Medical | 1269 (58.2%) | 832 (59.8%) |
| | Intensive care | 117 (5.4%) | 82 (5.9%) |
| Smoking status | Never | 1853 (85.0%) | 1151 (82.7%) |
| | Ever | 205 (9.4%) | 153 (11.0%) |
| | Current | 123 (5.6%) | 87 (6.3%) |
| Smoking number | Mean ± SD | 11.2 ± 5.3 (n = 123) | 10.5 ± 4.9 (n = 66) |
| Smoking year | Mean ± SD | 23.6 ± 10.0 (n = 123) | 22.9 ± 8.6 (n = 87) |
| SBP | Mean ± SD | 117.7 ± 14.8 (n = 2,181) | 114.7 ± 15.3 (n = 1,391) |
| DBP | Mean ± SD | 70.5 ± 10.7 (n = 2,181) | 68.3 ± 10.8 (n = 1,391) |
| BMI | Mean ± SD | 21.6 ± 2.8 (n = 2,181) | 22.5 ± 3.5 (n = 1,391) |
| Abdominal circumference | Mean ± SD | 77.7 ± 7.6 (n = 2,181) | 79.2 ± 9.1 (n = 1,391) |
| Eye test; right | Less than 0.6 | 398 (18.2%) | 266 (19.1%) |
| Eye test; left | Less than 0.6 | 400 (18.3%) | 275 (19.8%) |
| Hearing test; right 1 kHz | Abnormal | 44 (2.0%) | 27 (1.9%) |
| Hearing test; left 1 kHz | Abnormal | 40 (1.8%) | 16 (1.2%) |
| Hearing test; right 4 kHz | Abnormal | 28 (1.3%) | 21 (1.5%) |
| Hearing test; left 4 kHz | Abnormal | 21 (1.0%) | 20 (1.4%) |
| Medical examination | Some findings | 33 (1.5%) | 12 (0.9%) |
| ECG | Some findings | 414 (19.0%) | 298 (21.4%) |
| Chest X-ray | Some findings | 217 (9.9%) | 139 (10.0%) |
| History of DM treatment | Yes | 23 (1.1%) | 26 (1.9%) |
| History of HL treatment | Yes | 122 (5.6%) | 57 (4.1%) |
| History of HT treatment | Yes | 127 (5.8%) | 69 (5.0%) |
| New onset of MetS | Yes | 66 (3.0%) | 72 (5.2%) |

**Notes.**

SD, Standard Deviation; SBP, Systolic Blood Pressure; DBP, Diastolic Blood Pressure; BMI, Body Mass Index; ECG, Electro Cardiogram; DM, Diabetes; HL, Hyperlipidemia; HT, Hypertension; MetS, Metabolic Syndrome.

Table 4 shows the CATE differences between each category (department and sex). For the department, the medical division was used as the reference category. The CATE differences in the administrative, research, and intensive care departments were 0.002 (95% CI [−0.070–0.074]; $p = 0.951$), 0.036 (95% CI [−0.054–0.127]; $p = 0.434$), and 0.122 (95% CI [−0.054–0.298]; $p = 0.173$), respectively. For sex, males had a slightly higher CATE than females at 0.032 (95% CI [−0.049–0.114]; $p = 0.435$). However, none of the CATE differences were statistically significant.

Figure 5 shows a scatter plot and smoothing curves depicting the relationship among CATE, age, and department. Although the plot points are widely distributed, the smoothing

**Table 2  Descriptive statistics of test results at baseline (2).**

| | | Before COVID (N = 2,181) | After COVID (N = 1,391) |
|---|---|---|---|
| Urinary occult blood | (-) | 3 (0.1%) | 6 (0.4%) |
| | (+) | 1 (<1%) | 0 (0.0%) |
| | Missing | 2,177 (99.8%) | 1,385 (99.6%) |
| Urinary protein | (-) | 2,145 (98.3%) | 1,365 (98.1%) |
| | (±) | 21 (1.0%) | 9 (0.6%) |
| | (+) | 11 (0.5%) | 14 (1.0%) |
| | (2+) | 1 (<1%) | 2 (0.1%) |
| | (3+) | 2 (0.1%) | 0 (0.0%) |
| | Missing | 1 (<1%) | 1 (0.1%) |
| Urinary sugar | (-) | 2,157 (98.9%) | 1,371 (98.6%) |
| | (±) | 11 (0.5%) | 6 (0.4%) |
| | (+) | 6 (0.3%) | 7 (0.5%) |
| | (2+) | 6 (0.3%) | 2 (0.1%) |
| | (3+) | 0 (0.0%) | 2 (0.1%) |
| | (4+) | 0 (0.0%) | 2 (0.1%) |
| | Missing | 1 (<1%) | 1 (0.1%) |
| Hgb (g/dL) | Mean ± SD | 13.3 ± 1.5 (n = 2,181) | 13.7 ± 1.6 (n = 1,391) |
| RBC ($10^4$/μL) | Mean ± SD | 444.1 ± 41.3 (n = 2,181) | 456.4 ± 46.0 (n = 1,391) |
| AST (U/L) | Mean ± SD | 19.7 ± 6.2 (n = 2,181) | 20.8 ± 10.1 (n = 1,391) |
| ALT (U/L) | Mean ± SD | 16.9 ± 9.7 (n = 2,181) | 18.7 ± 15.0 (n = 1,391) |
| γ-GTP (U/L) | Mean ± SD | 29.0 ± 45.6 (n = 2,181) | 30.3 ± 37.0 (n = 1,391) |
| LDL Cholesterol (mg/dL) | Mean ± SD | 116.2 ± 29.0 (n = 2,181) | 120.5 ± 31.2 (n = 1,391) |
| HDL Cholesterol (mg/dL) | Mean ± SD | 67.7 ± 16.6 (n = 2,181) | 64.7 ± 17.4 (n = 1,391) |
| TG, fasting (mg/dL) | Mean ± SD | 86.8 ± 49.8 (n = 355) | 95.1 ± 53.9 (n = 347) |
| TG, at any time (mg/dL) | Mean ± SD | 100.8 ± 64.0 (n = 1,826) | 107.8 ± 88.1 (n = 1,044) |
| HbA1c (%) | Mean ± SD | 5.5 ± 0.3 (n = 1,056) | 5.6 ± 0.3 (n = 1,344) |
| BS, fasting (mg/dL) | Mean ± SD | 91.5 ± 12.2 (n = 216) | 90.7 ± 10.1 (n = 347) |
| BS, at any time (mg/dL) | Mean ± SD | 93.4 ± 18.4 (n = 909) | 93.0 ± 15.6 (n = 1,044) |

**Notes.**

SD, Standard Deviation; Hgb, Hemoglobin; RBC, Red Blood Cell; AST, Aspartate Aminotransferase; ALT, Alanine Aminotransferase; LDL, Low-density Lipoprotein; HDL, High-density Lipoprotein; TG, Triglyceride; HbA1c, Hemoglobin A1c; BS, Blood Sugar.

curves revealed a gradual increase in CATE with age. Furthermore, the intensive care department showed a higher CATE than the other departments at all ages.

## Characteristics of the high-CATE group

Tables 5 and 6 present the characteristics of the populations with the upper 10th percentile of CATE values (high) and the lower 10th percentile (low). The high group had a CATE of 0.261 or higher, and the low group had a CATE of −0.157 or lower.

The difference between the high- and low-CATE groups was especially large for the following items (standardized differences): age (0.766), history of treatment for hyperlipidemia (1.099), urinary protein (0.585), alanine aminotransferase levels (0.500),

**Table 3  CATEs and their 95% CI for the overall population as well as according to division and sex.**

|  | Number | CATE | 95% CI | | p-value |
| --- | --- | --- | --- | --- | --- |
| Total | 3,752 | 0.044 | 0.008 | 0.080 | 0.017 |
| Administrative division | 837 | 0.034 | −0.025 | 0.094 | 0.256 |
| Research division | 435 | 0.068 | −0.016 | 0.153 | 0.112 |
| Medical division | 2,101 | 0.032 | −0.012 | 0.077 | 0.158 |
| Intensive Care division | 199 | 0.154 | −0.021 | 0.329 | 0.084 |
| Female sex | 2,410 | 0.033 | 0.005 | 0.062 | 0.022 |
| Male sex | 1,162 | 0.066 | −0.016 | 0.148 | 0.116 |

**Notes.**
CATE, Conditional Average Treatment Effect; CI, Confident Interval.
95% CI was calculated by the bootstrap method.

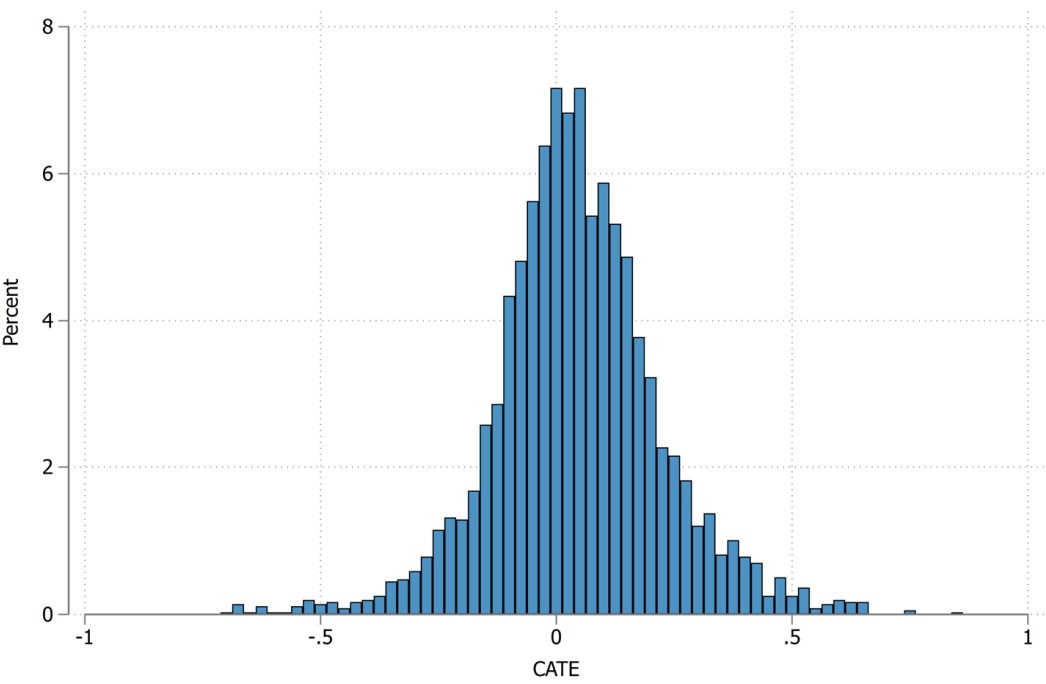

**Figure 4  Histogram of CATE distribution.** CATE, Conditional average treatment effect.

$\gamma$-glutamic pyruvic transaminase levels (0.611), fasting triglyceride levels (0.600), and hemoglobin A1c (0.501).

## Results based on the original dataset

There were no significant differences in the results after analysis of the original and anonymized datasets. CATE values calculated using the original dataset are presented in Tables S1 and S2 and Figs. S1 and S2.

**Table 4  Differences in CATE between categories and their 95% confidence intervals according to division and sex.**

|  | Number | Difference in CATE | 95% CI | | p-value |
|---|---|---|---|---|---|
| Administrative division | 837 | 0.002 | −0.070 | 0.074 | 0.951 |
| Research division | 435 | 0.036 | −0.054 | 0.127 | 0.434 |
| Medical division | 2,101 | 0 | (reference category) | | |
| Intensive Care division | 199 | 0.122 | −0.054 | 0.298 | 0.173 |
| Female sex | 2,410 | 0 | (reference category) | | |
| Male sex | 1,162 | 0.032 | −0.049 | 0.114 | 0.435 |

**Notes.**

CATE, Conditional Average Treatment Effect; CI, Confident Interval.

95% CI was calculated by the bootstrap method.

In division, medical division was used as the reference category, and in sex, female sex was used as the reference category.

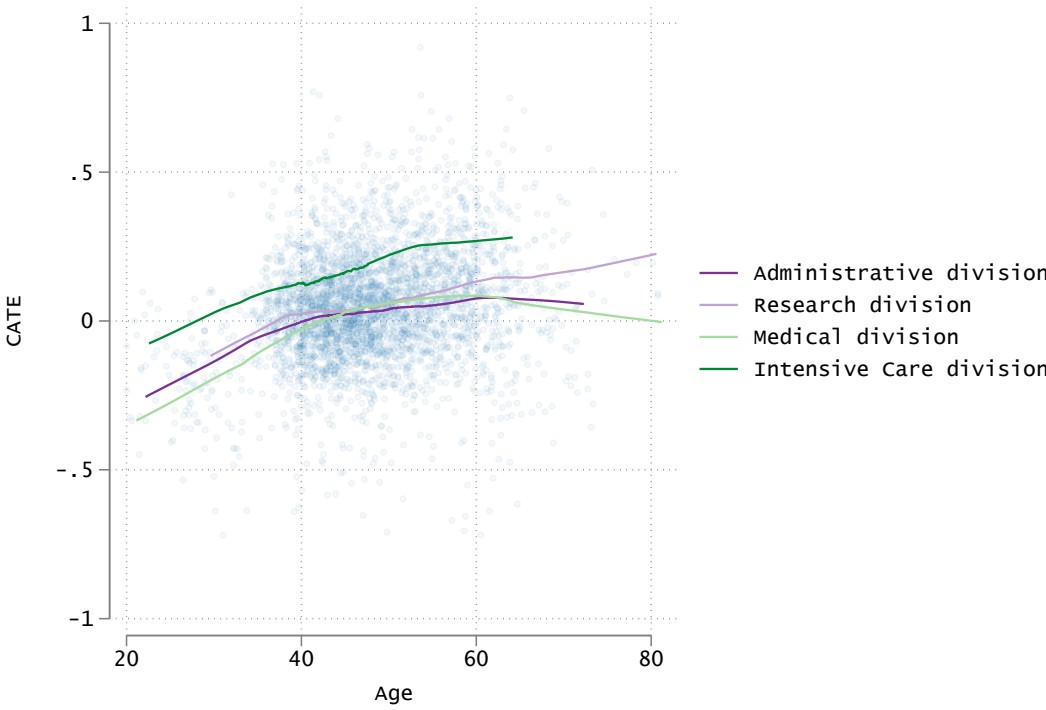

**Figure 5  Scatter plots of age and CATE and approximate curves by division.** CATE, Conditional average treatment effect.

## DISCUSSION

### Summary of the results

The COVID-19 pandemic increased the probability of new-onset MetS by 4.4%. This impact had heterogeneity; moreover, there was a larger impact in the intensive care department (15.4% increase) than in other departments. The high-CATE group was characterized by older age, urinary protein, elevated liver enzymes, higher triglyceride levels, increased hemoglobin A1c levels, and a history of treatment for hyperlipidemia.

**Table 5  Comparison of baseline characteristics between the low and high CATE groups (1).**

| | | Low CATE (N = 357) | High CATE (N = 357) | Std diff |
|---|---|---|---|---|
| Sex | Female | 219 (61.3%) | 169 (47.3%) | 0.28399 |
| | Male | 138 (38.7%) | 188 (52.7%) | |
| Age | Mean ± SD | 44.6 ± 11.4 | 52.0 ± 7.7 | 0.76620 |
| Division | Administrative | 92 (25.8%) | 67 (18.8%) | 0.44125 |
| | Research | 42 (11.8%) | 55 (15.4%) | |
| | Medical | 214 (59.9%) | 188 (52.7%) | |
| | Intensive Care | 9 (2.5%) | 47 (13.2%) | |
| Smoking status | Never | 267 (74.8%) | 270 (75.6%) | 0.11876 |
| | Ever | 56 (15.7%) | 44 (12.3%) | |
| | Current | 34 (9.5%) | 43 (12.0%) | |
| Smoking number | Mean ± SD | 2.5 ± 4.2 | 3.3 ± 5.8 | 0.14421 |
| Smoking year | Mean ± SD | 5.2 ± 9.1 | 6.0 ± 10.3 | 0.08440 |
| SBP | Mean ± SD | 119.9 ± 17.6 | 120.9 ± 13.2 | 0.06441 |
| DBP | Mean ± SD | 70.5 ± 12.3 | 73.3 ± 8.9 | 0.26378 |
| BMI | Mean ± SD | 23.1 ± 4.2 | 22.5 ± 2.7 | 0.16453 |
| Abdominal circumference | Mean ± SD | 80.3 ± 9.6 | 82.3 ± 8.1 | 0.22829 |
| Eye test; right | Less than 0.6 | 72 (20.2%) | 63 (17.6%) | 0.06442 |
| Eye test; left | Less than 0.6 | 70 (19.6%) | 57 (16.0%) | 0.09533 |
| Hearing test; right 1 kHz | Abnormal | 10 (2.8%) | 12 (3.4%) | 0.03242 |
| Hearing test; left 1 kHz | Abnormal | 6 (1.7%) | 9 (2.5%) | 0.05862 |
| Hearing test; right 4 kHz | Abnormal | 6 (1.7%) | 9 (2.5%) | 0.05862 |
| Hearing test; left 4 kHz | Abnormal | 2 (0.6%) | 11 (3.1%) | 0.18940 |
| Medical examination | Some findings | 4 (1.1%) | 8 (2.2%) | 0.08725 |
| ECG | Some findings | 80 (22.4%) | 56 (15.7%) | 0.17183 |
| Chest X-ray | Some findings | 60 (16.8%) | 22 (6.2%) | 0.33860 |
| History of DM treatment | Yes | 6 (1.7%) | 16 (4.5%) | 0.16263 |
| History of HL treatment | Yes | 4 (1.1%) | 143 (40.1%) | 1.09865 |
| History of HT treatment | Yes | 71 (19.9%) | 28 (7.8%) | 0.35395 |
| New onset of MetS | Yes | 20 (5.6%) | 41 (11.5%) | 0.21161 |

**Notes.**

CATE, Conditional Average Treatment Effect; Std Diff, Standardized Difference; SD, Standard Deviation; SBP, Systolic Blood Pressure; DBP, Diastolic Blood Pressure; BMI, Body Mass Index; ECG, Electrocardiogram; DM, Diabetes; HL, Hyperlipidemia; HT, Hypertension; MetS, Metabolic Syndrome.

## Interpretation of the results

In this study, the CATE differed according to age and sex. Older age was associated with higher CATE values; however, there was substantial interindividual variability. The prevalence of MetS is typically higher among elderly individuals (*Arai et al., 2006*; *Hirode & Wong, 2020*), which may explain their increased vulnerability to new-onset MetS during the pandemic. Moreover, the pre-pandemic prevalence of MetS was higher among males than among females (*Arai et al., 2006*; *Kudo et al., 2021*).

Furthermore, this study observed heterogeneity according to the department. The lower CATE in the healthcare department (excluding the intensive care department) may

**Table 6  Comparison of baseline characteristics between the low and high CATE groups (2).**

| | | Low CATE (N = 357) | High CATE (N = 357) | Std diff |
|---|---|---|---|---|
| Urinary occult blood | (-) | 1 (0.3%) | 2 (0.6%) | 0.08669 |
| | (+) | 0 (0.0%) | 1 (0.3%) | |
| | Missing | 356 (99.7%) | 354 (99.2%) | |
| Urinary protein | (-) | 300 (84.0%) | 355 (99.4%) | 0.58507 |
| | (±) | 28 (7.8%) | 0 (0.0%) | |
| | (+) | 22 (6.2%) | 2 (0.6%) | |
| | (2+) | 3 (0.8%) | 0 (0.0%) | |
| | (3+) | 2 (0.6%) | 0 (0.0%) | |
| | Missing | 2 (0.6%) | 0 (0.0%) | |
| Urinary sugar | (-) | 349 (97.8%) | 354 (99.2%) | 0.21067 |
| | (±) | 4 (1.1%) | 1 (0.3%) | |
| | (+) | 0 (0.0%) | 2 (0.6%) | |
| | (2+) | 2 (0.6%) | 0 (0.0%) | |
| | (3+) | 2 (0.6%) | 0 (0.0%) | |
| | (4+) | 13.5 ± 2.1 | 14.0 ± 1.3 | |
| | Missing | 446.8 ± 54.0 | 462.4 ± 44.7 | |
| Hgb (g/dL) | Mean ± SD | 19.6 ± 13.3 | 24.7 ± 11.7 | 0.27788 |
| RBC ($10^4$/µL) | Mean ± SD | 17.2 ± 11.7 | 25.9 ± 21.6 | 0.31614 |
| AST (U/L) | Mean ± SD | 23.8 ± 17.4 | 71.9 ± 109.8 | 0.41172 |
| ALT (U/L) | Mean ± SD | 117.3 ± 25.6 | 122.8 ± 34.8 | 0.50093 |
| $\gamma$-GTP (U/L) | Mean ± SD | 64.8 ± 17.8 | 62.3 ± 17.3 | 0.61145 |
| LDL Cholesterol (mg/dL) | Mean ± SD | 80.5 ± 13.2 | 103.2 ± 51.6 | 0.18067 |
| HDL Cholesterol (mg/dL) | Mean ± SD | 101.9 ± 64.7 | 110.7 ± 66.9 | 0.14462 |
| TG, fasting (mg/dL) | Mean ± SD | 5.5 ± 0.2 | 5.6 ± 0.4 | 0.59998 |
| TG, at any time (mg/dL) | Mean ± SD s | 89.8 ± 4.7 | 92.4 ± 9.4 | 0.13373 |
| HbA1c (%) | Mean ± SD | 89.2 ± 10.7 | 93.9 ± 14.9 | 0.50127 |
| BS, fasting (mg/dL) | Mean ± SD | 1 (0.3%) | 2 (0.6%) | 0.34506 |
| BS, at any time (mg/dL) | Mean ± SD | 0 (0.0%) | 1 (0.3%) | 0.35810 |

**Notes.**
CATE, Conditional Average Treatment Effect; Std Diff, Standardized Difference; SD, Standard Deviation; Hgb, Hemoglobin; RBC, Red Blood Cell; AST, Aspartate Aminotransferase; ALT, Alanine Aminotransferase; LDL, Low-density Lipoprotein; HDL, High-density Lipoprotein; TG, Triglyceride; HbA1c, Hemoglobin A1c; BS, Blood Sugar.

be attributed to several factors. For example, as healthcare workers are relatively more knowledgeable about health, they may have continued engaging in activities (*e.g.*, exercise and diet) to maintain their health during the COVID-19 pandemic. Additionally, some reports have suggested favorable behavioral changes among employed and highly educated individuals (*Knell et al., 2020*), which may explain some of the behavioral changes among healthcare workers.

Conversely, the CATE was higher among intensive care workers, which could be attributed to long working hours as well as high physical and psychological loads, including handling patients with COVID-19 (*Gupta & Sahoo, 2020*; *Koontalay et al., 2021*).

The high-CATE group was characterized by older age, urinary protein, elevated liver enzymes, higher triglyceride levels, increased hemoglobin A1c levels, and a history of hyperlipidemia treatment, which are indicative of suboptimal healthcare before the COVID-19 pandemic. Such individuals are more susceptible to new environmental stressors and may be at an increased risk of developing new-onset MetS.

Additionally, the marked difference in CATE values between sexes underscores the need for sex-specific prevention strategies, particularly in high-stress environments such as intensive care units. This indicates a nuanced interplay of biological and environmental factors contributing to MetS risk, which warrants further investigations to tailor preventive strategies effectively.

## Comparison with previous studies

This study showed both similarities and differences in the characteristics of workers whose health conditions are likely to deteriorate compared with previous studies on the adverse health effects of the COVID-19 pandemic on frontline healthcare workers, especially with respect to mental health (*Batra et al., 2020*; *Gupta & Sahoo, 2020*; *De Kock et al., 2021*; *Koontalay et al., 2021*). A previous study reported that female workers, workers with fewer enrollment years, and intensive care workers were at an increased risk of mental illness (*Azoulay et al., 2020*; *Matsuo et al., 2020*; *Zhu et al., 2020*; *Marvaldi et al., 2021*; *Smallwood et al., 2021*). Considering the results of previous and present studies, intensive care workers are observed to be more susceptible to both mental health issues and MetS. Notably, a sex-based disparity exists wherein females predominantly face mental health challenges, while males are more prone to MetS. Age also delineates this distinction; workers with fewer enrollment years (*i.e.,* many younger workers) tend to exhibit mental health concerns, whereas older individuals are more likely to develop MetS.

A history of mental health disorders has been found to predispose individuals to anxiety and depressive symptoms during the COVID-19 pandemic (*Zhu et al., 2020*; *Arpacioglu, Gurler & Cakiroglu, 2021*; *Smallwood et al., 2021*). In this study, the high-CATE group was characterized by a history of hyperlipidemia treatment and high laboratory values. Although the previous and present studies differ in diseases, both show that a history of pre-pandemic illness is associated with a poor health status during a major event such as a pandemic.

## Suggestions for prevention interventions

The present findings indicate the need for preventive interventions for the described high-CATE population during a pandemic. For example, it might be beneficial to inform this population regarding their health vulnerability during a pandemic. Moreover, strict treatment management should be thoroughly implemented for individuals with pre-existing conditions. During a pandemic, home exercises (*Chtourou et al., 2020*) should be encouraged and assistance provided to create an environment conducive for home exercises. *Pappa, Sakkas & Sakka (2022)* suggested the importance of enhancing social support networks to promote psychological resilience. Several interventions can be implemented to reduce social isolation during a pandemic, including psychotherapy,

friendship lessons, robot pets, social facilitation, coping strategies, and further education on the causative disease (*Williams et al., 2021*; *Gonçalves et al., 2022*). Another viable intervention for mitigating the isolation requirements of the COVID-19 pandemic is videoconferencing (*Williams et al., 2021*).

### Strengths and limitations of the study

This study has several strengths. First, this is the first study on the incidence of new-onset MetS among healthcare workers during the COVID-19 pandemic. Second, as this study used health checkup data, there were few misclassified data items. Third, the large sample size ($n = 3,572$) considerably reduced the influence of random errors. Fourth, the X-learner model allowed the flexible calculation of CATE, which captured the characteristics of the high-risk population that could not be identified by ATE alone.

Nevertheless, this study has some limitations. First, this was a single-center study, and generalizability of the study findings to other universities and hospitals cannot be assumed. Second, this study did not consider some predictor variables, including job type (doctors, nurses, and so on), working hours, stress levels, socioeconomic status (income and education level), physical activity, and social support. Incorporating these variables would allow the construction of a more accurate prediction model of base learners, which leads to less bias. However, some measured variables can serve as surrogate variables for unmeasured variables. By including surrogate variables in the statistical model, the impact of unmeasured variables will be minimized to some extent. For example, body mass index at baseline could be a surrogate variable for physical activity. Third, the exclusion of health checkup data of patients without MetS (4,098 unexposed, 1,892 exposed) could raise concerns regarding selection bias. However, the absence of MetS data was due to participants' younger age, as according to Japanese law, some MetS testing may be omitted for young workers. Thus, these exclusions are independent of exposure, and the resulting bias is expected to be minimal. Fourth, base learners M2 and M3 had limited performance. Instead of the beta regression model used in this study, a random forest regression model may allow an improved prediction accuracy. However, as the response variable in this study was a real number ranging from $-1$ to $+1$, random forest regression may also predict values outside this range. Furthermore, the same person was included in both the exposed and unexposed groups, which should generally be avoided with respect to independent and identically distributed variables. However, as indicated in the second paragraph of the Results section, the impact of this violation is considered limited. Finally, calculating the CI using the bootstrap method has been shown to result in narrow CIs for the X-learner estimates (*Künzel et al., 2019b*); therefore, the calculated CIs may not have 95% coverage.

## CONCLUSIONS

This study demonstrated an increase in the incidence of new-onset MetS during the COVID-19 pandemic in a single Japanese campus. Additionally, characteristics of the high-risk population, including working in the intensive care unit, were identified. High-risk populations require specific preventive interventions in case the current COVID-19

pandemic persists or a new pandemic occurs. Future studies should consider additional information such as working hours and data from stress checks and medical examinations.

## ACKNOWLEDGEMENTS

I would like to acknowledge the staff of Junpukai for their efforts in using the data. I would like to thank Editage for English language editing.

### Funding
This work was supported by the Occupational Health Promotion Foundation (Reiwa-3, No.142). The funders had no role in study design, data collection and analysis, decision to publish, or preparation of the manuscript.

### Grant Disclosures
The following grant information was disclosed by the author:
Occupational Health Promotion Foundation (Reiwa-3, No.142).

### Competing Interests
The authors declare there are no competing interests.

### Author Contributions
- Toshiharu Mitsuhashi conceived and designed the experiments, performed the experiments, analyzed the data, prepared figures and/or tables, authored or reviewed drafts of the article, and approved the final draft.

### Human Ethics
The following information was supplied relating to ethical approvals (*i.e.,* approving body and any reference numbers):

This study was ethically approved by the clinical research review committee of the Junpukai healthcare center (approval date: June 3, 2022, approval number: 20210015).

### Data Availability
The raw data is available in the Supplementary Files.

### Supplemental Information
Supplemental information for this article can be found online at http://dx.doi.org/10.7717/peerj.17013#supplemental-information.

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
