# Peer review of "Heterogeneity of the effect of the COVID-19 pandemic on the incidence of Metabolic Syndrome onset at a Japanese campus"

_PeerJ, doi:10.7717/peerj.17013_

## Round 0.1 · original submission · Major Revisions

Dear Dr. Mitsuhashi:

Thanks for submitting your manuscript to PeerJ. I have now received three independent reviews of your manuscript, and as you will see, the reviewers raised some concerns about the research. Despite this, these reviewers are very optimistic about your work and the potential impact it will have on research studying COVID-19 and metabolic syndrome. Thus, I encourage you to revise your manuscript, accordingly, considering all the concerns raised by the reviewers.

There are many concerns pointed out by the reviewers, and you will need to address all of these and expect a thorough review of your revised manuscript by these same reviewers.

Please be sure to clarify your methods and statistics and make sure the workflow is repeatable.

I look forward to seeing your revision, and thanks again for submitting your work to PeerJ.

Good luck with your revision,

Best,

-joe

Reviewer 1 ·

Basic reporting

1. In the "Statistical analysis" section of the manuscript, it is recommended that the author provides a more comprehensive explanation of the term "conditional average treatment effect" (CATE) both mathematically and in language, and possibly comparing it to the average treatment effect (ATE). This clarification will enhance the clarity of the paper and reduce any potential for confusion among readers.

2. Within the same section, the author has referenced a limited number of papers that are specifically related to the X-learner. To provide a more well-rounded understanding of CATE, it is advisable for the author to include additional citations that address the broader concept of general conditional average treatment effects (CATE).

Experimental design

1. The author should clarify the assumptions used in the study. For instance, the division of data into controlled and treated subsets based on the year as the exposure should be thoroughly explained and justified. This could involve referencing similar studies with a comparable data structure.

2. Additionally, when stating that "the same person was included in both the exposed and unexposed groups...However,...this limitation did not significantly affect the results..." (lines 322-326), it would be more convincing to provide the specific number of individuals in both groups. This is important because, in extreme cases, it's possible that all subjects are part of both groups, making both potential outcomes observable.

Validity of the findings

1. In this study, there are several limitations that need to be addressed. Firstly, the data is derived solely from workers at a single campus, which makes it challenging to generalize the findings to other groups. Consequently, the author's conclusion and the title appear to be overly broad and generalized, given the limited scope of the study.
2. One specific concern is related to how the author defines the high-CATE group, which involves selecting the upper 10th percentile and comparing it to the lower 90th percentile group. To enhance the study's validity, it is advisable for the author to consider a direct comparison between the upper 10th percentile group and the lower 10th percentile group. With this revised approach, it would be expected that more significant standard differences would emerge in characteristics like age and abdominal circumference between the high-CATE and low-CATE groups, should the author's findings hold true.
3. Furthermore, the author draws attention to individuals working in the intensive care unit as a high-risk population, primarily based on p-values testing if the causal effect is equal to zero. A more common and reasonable approach would be to compare the causal effects between the intensive care unit group and other groups to determine if they are indeed higher. It is worth noting that the sample size for the intensive care unit group is relatively small, which introduces a higher level of uncertainty. While the p-value for the intensive care unit group may be the lowest, the research division achieved a similar p-value (with a difference of only 0.028). The author should acknowledge and address this issue in the study.
4. In the 'Evaluation of Base Learners' section, RMSE is presented as a metric for model performance. However, RMSE, in isolation, lacks interpretability as there is no established benchmark for determining whether a particular RMSE value is considered low or high. Also, it would be beneficial to include the interpretation of AUC for the first-stage models.

Reviewer 2 ·

Basic reporting

Firstly, we would like to commend the authors for their expansive dataset. Furthermore, the article under review includes a significant proportion of women, effectively addressing the underrepresentation of female participants in scientific literature.

The article maintains professional use of the English language and a conventional, yet effective, professional structure. The authors' inclusion of relevant literature references provides sufficient context for their research. The authors' willingness to share raw data enhances the article's transparency. The research remains self-contained, presenting results that directly align with the stated hypotheses.

We would like to point out some suggestions for improvement:

1.- Lines 17 and 57: Substitute the Word “curb” for a more appropriate word in this context such as “reduce” or “slow” the spread of the virus.

2.- Line 22: Replace the word "inform" with "direct," which is more suitable for the context.
Line 28 and 225: Substitute the Word “Further” for a more appropriate word in this context such as “Furthermore”.

3.- Line 35-36: To indicate numbers, please use commas, as follows: 3,572; 2,181; or 1,544.

4.- Line 86: Replace “to my knowledge” for the phrase “to the best of our knowledge” to employ the “royal we” or “majestic plural” although the article is written by a single author.

5.- Line 93: the name of the study design that utilizes previously collected data is called “observational retrospective study”. Please include this in your manuscript.

6.- Line 104: The current phrasing makes comprehension difficult. Please change “those with only data other than general health checkups” for “individuals who only had data unrelated to general health assessments”

7.- Line 120: The URL provided with the diagnostic criteria of metabolic syndrome is in Japanese. Please provide the English version of the document to facilitate comprehension for the general audience of the journal. Furthermore, to the best of our knowledge there are different diagnostic criteria used to define metabolic syndrome, such as the AHA/NHLBI (2009), the IDF (2006) or the ATPIII (2001) criteria (references below). It would be interesting to add a short parragraph explaining how these criteria compare to the diagnostic criteria of the Ministery of health, labour and welfare of Japan to these well known definition.
a) AHA/NHLBI Saklayen, M.G. The global epidemic of the metabolic syndrome. Curr. Hypertens. Rep. 2018, 20, 1–8
b) IDF: Eckel, R.H.; Alberti, K.G.; Grundy, S.M.; Zimmet, P.Z. The metabolic syndrome. Lancet 2010, 375, 181–183
c) ATPIII: Expert Panel on Detection, Evaluation, and Treatment of High Blood Cholesterol in Adults. JAMA 2001, 285, 2486–2497

8.- Line 128: Kindly clarify the variables encompassed within the "health-checkup items." If there is an extensive list that cannot be explicitly mentioned in the text, please direct readers to a table for reference.

Experimental design

The submitted article, aligns with the journal's scope. It poses a well-defined and relevant research question, effectively identifying a knowledge gap and highlighting how the study fills it. The research maintains ethical standards, and the method is described in detail to facilitate replication.

1.- Identifying potential confounding variables in this study is crucial to ensure the accuracy and validity of the results. Please state if you have controled for other confounding variables such as: socioeconomic status (income and education level), physical activity, social support...etc.

Validity of the findings

The conclusions are aptly formulated, directly tied to the initial research query. The availability of underlying data is a commendable aspect of the study.

Additional comments

The section titled "Comparison with previous studies" (lines 273-287) appears somewhat unclear, especially in the context of mental health. It would greatly enhance the discussion if you could provide information about the mental health status of your study participants and the differences between the exposed and unexposed groups, such as data from validated surveys, occurrences of new-onset mental health issues, or referrals to mental health specialists during the pandemic. While it is pertinent to discuss the potential influence of the COVID-19 pandemic on mental health and its relationship with the onset of metabolic syndrome, drawing inferences without specific participant data might be a bit of a stretch. For instance, asserting that patients in the high CATE group had a history of hyperlipidemia, which is linked to poorer mental health and, consequently, new-onset metabolic syndrome, could be seen as speculative without concrete, participant-specific data.

Reviewer 3 ·

Basic reporting

no comment

Experimental design

no comment

Validity of the findings

no comment

Additional comments

In this study, author aimed to assess the heterogeneity of the impact of the COVID-19 pandemic on the incidence of new-onset metabolic syndrome (MetS) among university campus workers. This topic is interesting however, there are some important observations that prevent me from accepting this paper.

1- In the introduction section, it is essential to include a paragraph that provides background information about metabolic syndrome (MetS) and explicitly highlights the motivation behind conducting this study to assess the heterogeneity of the COVID-19 pandemic's impact on the incidence of new-onset MetS. In summary, adding a paragraph on metabolic syndrome and the rationale for studying its relationship with the COVID-19 pandemic will provide readers with a clear understanding of the study's purpose and significance.

2- The criteria by which the observations were diagnosed as metabolic syndrome should be explained in detail in the materials methods section.
“Follow-up data were used to determine whether participants had MetS based on the diagnostic criteria provided by the Ministry of Health, Labour and Welfare (short URL: https://bit.ly/3Jx9b8g).” This information is insufficient to assess individuals' metabolic syndrome status and should be further elaborated for clarity.

3- A more detailed discussion section would enhance the interpretation of results and their practical implications.

4- The base learners were assessed for their predictive utility using logistic and beta regression models. Please provide more details about usage of logistic and beta regression models for evaluation of base learners

5- To my knowledge, MtS is defined according to the National Cholesterol Education Program (NCEP) Adult Treatment Panel III (ATP III) identification [69]. Based on this description, MtS is present if three or more of the following five criteria are provided: Waist circumference over 102 cm (male) or 88 cm (female) as high central fat accumulation, blood pressure (SBP/DBP) over 130/85 mmHg, fasting Triglycerides level over 150 mg/dl, fasting HDL cholesterol level less than 40 mg/dl (men) or 50 mg/dl (women) and fasting blood sugar over 100 mg/ dl.
However, in your text, despite the importance of metabolic syndrome as a criterion, I couldn't find a detailed paragraph regarding this information.

7- What sampling method did you use in your study? Please specify it.

---

## Round 0.2 · Minor Revisions

Dear Dr. Mitsuhashi:

Thanks for revising your manuscript. The reviewers are very satisfied with your revision (as am I). Great! However, there are a few minor issues to attend to. Please address these ASAP so we may move towards acceptance of your work.

Best,

-joe

Reviewer 1 ·

Basic reporting

1. The title could be "...a Japanese campus" by removing "single," but this is optional.

Experimental design

1. The author's definition of the control and exposure groups requires additional assumptions, especially given the time-series nature of the data. One implicit assumption is that the potential outcomes, given the covariates before and after COVID-19, are the same. There may be other implicit assumptions in the study. The author should clarify how the proposed method allows for the estimation of the causal estimand.

2. Based on the numbers presented by the author, it appears that nearly 5 out of 6 subjects are sampled repeatedly. This substantial proportion raises concerns about the independence assumption, which seems to be severely violated. The author should highlight this issue earlier in the paper rather than addressing it towards the end.

Validity of the findings

1. As mentioned earlier, the violation of the independence assumption is a concern. It is recommended that the author assess the robustness of the findings against this dependency. This could involve conducting a formal sensitivity analysis or running simulations under a similar setting to gauge the impact of the dependency on the results.

Reviewer 2 ·

Basic reporting

We have identified an error in page 7, line 152. There is a word missing: cardiovascular risk "factors".

Experimental design

No comment.

Validity of the findings

No comment.

Additional comments

After carefully reviewing the revised manuscript, we have verified that the authors have diligently corrected the language errors and comprehensively addressed the concerns we previously highlighted. We believe these modifications contribute significantly to strengthening the manuscript's impact and readability. Following this review, we have no further comments to add.

Reviewer 3 ·

Basic reporting

After the revision, the revised manuscript is acceptable.

Experimental design

no comment

Validity of the findings

no comment

Additional comments

no comment

---

## Round 0.3 · accepted · Accept

Dear Dr. Mitsuhashi:

Thanks for revising your manuscript based on the concerns raised by the reviewers. I now believe that your manuscript is suitable for publication. Congratulations! I look forward to seeing this work in print, and I anticipate it being an important resource for groups studying COVID-19 and metabolic syndrome. Thanks again for choosing PeerJ to publish such important work.

Best,

-joe